# High Prevalence of *Strongyloides* among South Asian Migrants in Primary Care―Associations with Eosinophilia and Gastrointestinal Symptoms

**DOI:** 10.3390/pathogens9020103

**Published:** 2020-02-06

**Authors:** Elinor Chloe Baker, Damien K. Ming, Yasmin Choudhury, Shahedur Rahman, Philip J. Smith, Jose Muñoz, Peter L. Chiodini, Chris J. Griffiths, Christopher J. M. Whitty, Michael Brown

**Affiliations:** 1Whipps Cross University Hospital, London E11 1NR, UK; ecbaker@doctors.org.uk; 2Hospital for Tropical Diseases, University College London Hospitals, London WC1E 6JB, UK; p.chiodini@nhs.net (P.L.C.); christopher.whitty@lshtm.ac.uk (C.J.M.W.); michael.brown@lshtm.ac.uk (M.B.); 3William Harvey Heart Centre, Bart’s and the London School of Medicine and Dentistry, Queen Mary University of London, London EC1M 6BQ, UK; yasmin.choudhury@qmul.ac.uk; 4Riverside Medical Centre, Hockley SS5 6JY, UK; shahedurrahman@nhs.net; 5Royal Liverpool University Hospital, Liverpool L7 8XP, UK; drphilipjsmithbsg@gmail.com; 6Barcelona Centre for International Health Research, 08014 Barcelona, Spain; jose.munoz@isglobal.org; 7Institute of Population Health Sciences, Barts and the London School of Medicine and Dentistry, Queen Mary University of London, London E1 2AT, UK; c.j.griffiths@qmul.ac.uk; 8Clinical Research Department, London School of Hygiene & Tropical Medicine, London WC1E 7HT, UK

**Keywords:** *Strongyloides*, migrant health, primary care, gastrointestinal symptoms, eosinophilia, ivermectin

## Abstract

Gastrointestinal (GI) symptoms are a frequent reason for primary care consultation, and common amongst patients with strongyloidiasis. We conducted a prospective cohort and nested case control study in East London to examine the predictive value of a raised eosinophil count or of GI symptoms, for *Strongyloides* infection in South Asian migrants. We included 503 patients in the final analyses and all underwent a standardised GI symptom questionnaire, eosinophil count and *Strongyloides* serology testing. Positive *Strongyloides* serology was found in 33.6% in the eosinophilia cohort against 12.5% in the phlebotomy controls, with adjusted odds ratio of 3.54 (95% CI 1.88–6.67). In the GI symptoms cohort, 16.4% were seropositive but this was not significantly different compared with controls, nor were there associations between particular symptoms and Strongyloidiasis. Almost a third (35/115) of patients with a positive *Strongyloides* serology did not have eosinophilia at time of testing. Median eosinophil count declined post-treatment from 0.5 cells × 10^9^/L (IQR 0.3–0.7) to 0.3 (0.1–0.5), *p* < 0.001. We conclude *Strongyloides* infection is common in this setting, and the true symptom burden remains unclear. Availability of ivermectin in primary care would improve access to treatment. Further work should clarify cost-effectiveness of screening strategies for *Strongyloides* infection in UK migrant populations.

## 1. Introduction

*Strongyloides stercoralis* is a gastrointestinal helminth infection endemic in most tropical and subtropical regions, with an estimated 350 million people infected worldwide [1,2]. Prevalence among migrants in European countries is also thought to be high, ranging from 2% to 46% depending on the technique used (microscopy vs. serology) and the study population, though the infection is rarely diagnosed in non-specialized health centres [3,4]. The adult worm resides in the small intestine, generating progeny which are excreted in the faeces to complete the life cycle in the environment, but may develop into infective filariform larvae prior to defecation, penetrating the intestinal wall [5]. This unique autoinfection cycle allows strongyloidiasis to become a chronic infection that can persist lifelong in the infected individual, if not treated [6].

Health consequences of chronic infection are most severe in cases of immunosuppression, which may be iatrogenic, when unmoderated parasitic replication can lead to hyperinfection syndrome and often fatal disseminated strongyloidiasis [7]. Early diagnosis and treatment is effective at preventing severe disease in high risk individuals [8]. Indeed, economic analyses conducted in the US have shown that screening for eosinophilia and presumptive antiparasitic treatment were cost-effective strategies to prevent these complications in migrants [9,10]. However, these studies predated current approaches to diagnosis and treatment and a more targeted approach may be appropriate in some settings.

Chronic infection in immunocompetent hosts is commonly asymptomatic, and often eosinophilia may be the only clinical marker of infection [11]. However, in cohorts of patients with chronic infection, prevalent symptoms may include gastrointestinal complaints such as epigastric and other abdominal pain, diarrhoea and nausea [12]. These common symptoms may be overlooked in primary care as they are non-specific and associated with functional disorders such as Irritable Bowel Syndrome (IBS). Some such studies suggest an attributable (and easily reversible) burden of nonspecific GI symptoms related to intestinal helminths in secondary care. These symptoms present a health resource burden in primary care [13].

We explored the prevalence of strongyloidiasis among patients attending inner London primary care clinics serving a predominantly South Asian population, to establish the predictive value of incidentally raised eosinophil count, and of nonspecific GI symptoms, for identifying infection in this setting. We employed standardised questionnaires pre- and post-treatment to explore associations between symptoms and *Strongyloides* infection and treatment

## 2. Materials and Methods

This was a prospective cohort study with a nested case-control element and took place in two primary care practices in the Borough of Tower Hamlets, East London. In these practices, 45%–90% of the patient population of Bangladeshi heritage. This setting was chosen because of high *Strongyloides* referral rates into our practice from this migrant population [14].

Recruitment was restricted to migrants from *Strongyloides*-endemic countries, attending primary care for a general practitioner (GP) consultation.

Three groups were identified for recruitment:Patients who attended for non-specific gastrointestinal symptoms (with which they had presented at least twice over a six-month period and without diagnosis or treatment).Patients whose most recent full blood count (within last three months) demonstrated eosinophilia (defined as equal to or greater than 0.5 cells × 10^9^/L).Patients attending the in-practice phlebotomy service for a full blood count, for any reason. These patients formed the control group.

The study recruitment period was from August 2011 to December 2012. Recruitment of patients with eosinophilia from their most recent full blood count screening and GI symptoms was concurrent, control recruitment was temporally separated. Participants were identified using weekly systematic searches of the GP records system. They were invited to participate in the study by their GP, phlebotomist or by a letter (in the appropriate language) from the research team. After providing information, confirming that they were born in a *Strongyloides*-endemic country, and signing informed consent, a face to face structured interview was conducted by a member of the research team; either directly in Sylheti (a common language spoken by the Bangladeshi community) or with a translator. Epidemiological data (age, sex, country of birth, history of recent travel to endemic areas, number of years living in UK) and clinical data (HIV, immunosuppressive drugs―defined as immunomodulatory drugs including oral corticosteroid use more than 2 weeks in duration, history of helminth infection, previous GI diagnosis and history of atopy) were recorded during the interview. Patients who did not attend follow-up would be contacted through a Sylheti speaker on the telephone and another appointment arranged where possible.

Specific questions about gastrointestinal symptoms were asked to all participants. A first set of questions were included after adapting a clinical questionnaire [15], aimed at describing symptoms in patients with irritable bowel syndrome (IBS). This questionnaire included a detailed description of whether the patient had abdominal pain, abdominal distension, degree of satisfaction after defecation (tenesmus), and general impact of these symptoms on quality of life. A composite score (IBS score) was then calculated, and then patients classified as “No IBS” (0–7), “mild IBS” (75–175), “Moderate IBS” (175–300) and “Severe IBS” (>300). A second group of clinical questions was included after adapting the Leeds Dyspepsia Questionnaire (LDQ) that was designed to assess dyspepsia symptoms in the Asian population [16]. 

Serological samples were obtained by the practice phlebotomy staff or the research physician, and a routine full blood count was conducted for those who had not had one in the last 3 months. *Strongyloides* serology samples were transported to the national reference laboratory based at the Hospital for Tropical Diseases (HTD), University College London Hospital (UCLH), London, by the research team for next day analysis. *Strongyloides ratti* serology (Bordier affinity products, EC reg. N°: H-CH/CA01/IVD/10285) was used. Results were reported as optical density values for the test and assay reference sample. A ratio over 1 was considered a positive serology. A fresh stool sample was requested from all participants with positive serology and was analysed in the HTD laboratory. Microscopy of stool was performed on formol-ether concentration preparation, together with charcoal culture to increase sensitivity of parasitological methods. Stool analysis was restricted to those with positive serology for practical reasons.

Patients with positive *Strongyloides* serology were sent an appointment letter and were contacted by telephone to invite them to the Hospital for Tropical Diseases to initiate treatment. Those not attending were re-invited twice. Participants were invited for follow-up at HTD six months after treatment, and those who attended the visit underwent repeat serology and full blood count. A repeat questionnaire was also completed by the research team via telephone six months after treatment, or refusal of treatment, for all patients with positive serology. The study was approved by the NHS national research ethics committee (study no. 09/H0701/66).

## 3. Statistical Analysis

Case-control analyses were performed for each group, calculating odds ratios and confidence intervals for potential risk factors. Univariate analyses were conducted using Mann-Whitney and Chi-square tests; Fisher’s Exact test was employed when small cell sizes were found (values less than 5). Multivariate analyses were adjusted for age, sex, country of origin and time spent outside a *Strongyloides*-endemic area. The study was designed to have 80% power to detect a difference in prevalence of 20% vs. 10% between groups (e.g., eosinophilia vs. GI symptoms). All analyses were performed in Stata 14 (StataCorp, Texas).

## 4. Results

A total of 635 patients were identified in the electronic record system as having a raised eosinophil count or presenting with GI symptoms, and invited to join the study. Three-hundred and thirty-five patients were identified to have eosinophilia, of whom 224 (67%) were successfully recruited; 275 were found to have chronic GI symptoms, of whom 167 (61%) were successfully recruited; and 25 patients were identified as having both GI symptoms and eosinophilia, of whom 19 (76%) were recruited and included in the GI symptom arm of the study. One-hundred and twenty-two phlebotomy controls were recruited. 

### 4.1. Baseline Results

A total of 532 patients were recruited to the study. Twenty-nine patients were excluded from the final analysis because of incomplete data or withdrawal from the study therefore 503 patients were included in the final analysis―these results are displayed in Table 1. The median age was 49 years old (IQR 38–61), 55% were female and the most common countries of origin were Bangladesh (90.1%), Somalia (2.6%) and India (2.2%). There were fewer women in the eosinophil group than in the GI group or control group, and persons of Bangladeshi heritage made up a greater proportion of the control group than either the eosinophil or GI symptom groups; other socio-demographic characteristics were similar. 

Questionnaire responses demonstrated that those recruited to the GI symptom group (i.e., those for whom the reason for attendance was recorded as a GI symptom) were more likely to have GI symptoms, that impacted more severely on their lives, and were more likely to have undergone GI investigations than those from the other two groups. However, there was also a significantly higher median burden on quality of life in patients in the eosinophilia cohort when compared with the controls: (Median score (IQR), 0 (0–35) vs. 0 (0–4), *p* = 0.001). Patients within the eosinophilia cohort were also associated with a higher GI symptomatology on questioning compared with controls (Median score (IQR), 54 (0–160) vs. 10 (0–104), *p* = 0.002). There was also a possible impact on work: 84.5% of those with eosinophilia did not miss any work compared with 93.6% without eosinophilia, *p* = 0.03. 

A total of 14 patients recruited had received immunosuppressive medications over the previous five years of which 3/14 (21%) had positive *Strongyloides* serology. 

### 4.2. Analyses of Associations with Strongyloides Diagnosis

A total of 115 patients (23%) were found to have positive *Strongyloides* serology. Median eosinophil counts in *Strongyloides* positive patients were significantly higher compared with seronegative patients (0.5 cells × 10^9^/L (IQR 0.3–0.7) vs. 0.4 (IQR 0.2–0.5), *p* < 0.001). The overall positive and negative predictive values of eosinophilia for a positive *Strongyloides* serology across all three groups were 32% and 86% respectively, and 25% and 89% respectively for the control group. Thirty-five patients (30%) with positive *Strongyloides* serology had normal eosinophil counts. Of this group 11/35 patient underwent stool microscopy/culture and one patient had microbiologically-proven Strongyloidiasis.

Prevalence of *Strongyloides* was significantly higher amongst patients recruited for eosinophilia (33.6%, *p* < 0.001) than those presenting with GI symptoms (16.4%) or phlebotomy controls (12.5%). The difference in prevalence between the latter groups was not statistically significant. The odds ratios of patients with eosinophilia and positive *Strongyloides* serology was 3.55 (95% CI: 1.93–6.52) and remained significant after adjustment for possible confounders―adjusted odds ratios 3.54 (95% CI: 1.88–6.67). Questionnaire responses demonstrated no specific symptoms that were more common among *Strongyloides* positive patients (Table 2), and infection was not associated with time missed from work, mean IBS scores, or impact on quality of life scores. After adjustment for possible confounders there were no significant differences in time missed from work or impact on quality of life in multivariate analysis. Additionally, there were no differences in median age, gender, country of birth or median duration spent in the UK between the two groups. We also explored possible associations between IBS subtypes (IBS-diarrhoea, IBS-constipation and mixed) with *Strongyloides* serology and re-classified IBS subtypes according to criteria based on modified Rome IV classification [17], but were unable to find any significant associations. 

### 4.3. Treatment and Follow-up

Of the 115 *Strongyloides* positive patients through serology, 87 (76%) were seen at the HTD and received treatment with 200 μg/kg of ivermectin. The remaining patients either left their GP practices before the referral could be made, declined referral or did not attend despite multiple attempts to arrange an appointment. Only 38 (33% of all positive patients, 44% those attending HTD) *Strongyloides* positive patients provided a stool sample at their HTD clinic visit for which there was sufficient material for charcoal culture in 24 patients. A total of 9 (23%) were positive on microscopy or culture.

Follow up eosinophil count (at least 6 months after treatment or recruitment) was available for 47 patients who had a positive serology and received treatment. Within this group, the median eosinophil count fell from 0.5 cells × 10^9^/L (IQR 0.3–0.7) to 0.3 (IQR 0.1–0.5), *p* < 0.001 (Figure 1), and the median IBS score also reduced from 104 (IQR 0–224) to 15 (IQR 0–100), *p* = 0.004 after treatment.

### 4.4. Associations between Treatment and GI Symptoms

A total of 66 patients attending HTD for treatment had complete GI questionnaires before and 6 months after treatment: 17/29 (59%) with abdominal pain, 20/31 (65%) with abdominal distension, 8/12 (67%) with nausea, 12/27 (44%) with indigestion, 7/12 (58%) with diarrhoea and 10/26 (38%) with constipation reported resolution of symptoms in their follow-up questionnaire.

A further 12 patients who did not attend for treatment had complete GI questionnaires at diagnosis and at 6 months. Then, 2/12 (17%) reported overall improvement in their GI symptoms compared with 35% those who did attend for treatment (*p* = 0.23).

## 5. Discussion

This prospective, primary care-based study with a nested-case control analysis has demonstrated a high prevalence (12.5%) of strongyloidiasis amongst a control group of patients referred to the practice phlebotomist for a full blood count (for any reason) in GP practices in East London. Prevalence of a positive *Strongyloides* serology was 33.6% among patients with eosinophilia, and 16.4% among patients presenting to primary care with gastrointestinal symptoms. We infer that this seroprevalence of the control group further applies to the local migrant population, and this estimate is consistent with recent systematic reviews where pooled strongyloidiasis seroprevalence of migrants was estimated to be 12.2% [2]. While eosinophilia, a readily accessible result from a standard full blood count, is clearly a predictor of infection, the similar prevalence among controls and GI patients, and the lack of association between GI symptoms and *Strongyloides* infection in the questionnaires, precludes substantial conclusions about the association between nonspecific GI symptoms and chronic *Strongyloides* infection. This finding accords with existing literature which has been inconsistent in demonstrating associations between GI symptoms and chronic *Strongyloides* infection, and where asymptomatic chronic infection is common [18]. 

It is important to note that almost a third of patients (35/115, 30%) with a positive *Strongyloides* serology in our study did not have eosinophilia at time of testing and the overall negative predictive value was 86%. Eosinophilia is recognised to be transient [19] and therefore an effective screening strategy has to take into account individual risk. Patients within the eosinophilia cohort also had a significantly lower reported quality of life and increased GI symptomatology when compared with controls. The reasons for this are unclear but could relate to a multitude of factors, including other underlying pathologies which give rise to eosinophilia which have yet to be diagnosed. We found that 3/14 (21%) patients in our cohort who had underwent immunosuppression in the past five years also had a positive *Strongyloides* serology―this is an important clinical finding, with potentially catastrophic consequences if left untreated because of risk of hyperinfection. Furthermore, the sensitivity of serology can be attenuated in immunosuppression and stool microbiology should be performed in conjunction in these situations [20]. Screening through serology in a situation with a high proportion of patients with immunosuppression will therefore contribute to an under-estimation of overall disease prevalence.

We were unable to demonstrate any predictors of strongyloidiasis beyond eosinophilia. The lack of association with duration in the UK is to be expected with this helminth, which has an autoinfection cycle supporting persistent infection and for which effective treatments (ivermectin, or even albendazole) are not available in primary care settings, and investigation in secondary care infrequent [12]. We would be in favour of developing treatment algorithms for use in primary care which focus on patients who have travelled to *Strongyloides*-endemic regions [21] and present with an unexplained eosinophilia. In a case series with a similar underlying population from South Asia, presence of diabetes was associated with strongyloidiasis in multivariate analyses and this might form the basis of targeted screening strategies [22]. Further work should clarify the cost-effectiveness of screening strategies based on full blood count or serology for *Strongyloides* infection specific to UK migrant populations, as such analyses have been based on data conducted elsewhere and may not apply [8]. 

There was a high prevalence of nonspecific gastrointestinal symptoms in all groups. Indeed, Irritable Bowel Syndrome (IBS) is increasingly recognised to be a common disorder (with incidence as high as 70/1000 patient years), presenting frequently to primary care [23]. Although our data do not support specific associations of gastrointestinal symptoms with *Strongyloides* infection, seroprevalence of positive *Strongyloides* antibodies among patients with GI symptoms was reasonably high (16.4%) and the potential role of screening for a gastrointestinal parasite in these patients, as part of a management algorithm, remains attractive. There was limited data to suggest diagnosis and treatment was effective in improving symptoms based on our incomplete follow up and small sample size. Additionally, it is important to note that the assessment tools used to examine IBS severity or GI symptomatology in our study remains subjective which may restrict comparability between patients. A composite IBS severity score was utilized in this study, whereas a more precise classification such as one based on the 2016 Rome IV classification [17] may yield individual phenotypes with different associations to strongyloidiasis. We further explored this relationship with an IBS classification in light of the Rome IV classification which separates patients into predominant diarrhoea or constipation subtypes, but were unable to demonstrate any significant association. We acknowledge that this approach has limitations and it is unclear to what extent this classification is indeed applicable to our migrant population. 

The study had several limitations. Firstly, the control group constituted a slightly different demographic from the other arms with fewer persons of non-Bangladeshi heritage and fewer males. There may have been an underrepresentation of eosinophilia in this group compared to the practice as a whole. *Strongyloides* serology was also used rather than microscopy for diagnosis of chronic infection. This was principally for pragmatic reasons: recruitment requiring provision of stool samples in a primary care setting was considered highly unlikely to be successful and it is known that stool-shedding is intermittent and insensitive. There are data to suggest that serology testing is effective in the migrant population with sensitivities ranging from 90% to 98% in our experience when compared with microbiological testing [14,24]. We acknowledge that the low rate of stool investigations in those with a positive *Strongyloides* serology does not exclude the possibility of co-infection with other helminths in turn causing eosinophilia. It is well-recognised that *Strongyloides* serology may cross-react with filarial infection and other nematodes [25]. However, those diagnosed with positive *Strongyloides* serology had no clinical or epidemiological features to suggest filarial infection, and the majority of patients had a resolution of a previously sustained eosinophilia, in line with previous studies. We assumed that characteristics of the control group were similar to that of the underlying migrant population and therefore seroprevalence values would apply to the community. We did observe an improvement in symptoms in our follow up cohort and this might be of practical use to GPs in managing patients with these symptoms. However, given improvement in symptoms on follow up was in line with that seen in placebo arms of many interventions examining IBS over time [26], we recognize the limitations of these results.

There was a significant loss to follow up of patients after diagnosis of strongyloidiasis, in spite of study conditions: a native Sylheti research assistant provided clear directions to the HTD. A proportion of patients declined to attend for treatment and/or were not contactable despite repeated attempts. In addition, only a low (44%) proportion of patients who attended the HTD subsequently provided stool sample, limiting our ability to confirm infection. Many of the patients in our cohort migrated to the UK in the 1970s and remain highly mobile, often working long hours [27]. These patients may not be able to access healthcare effectively because of complex socioeconomic, as well as linguistic and cultural factors [28]. Findings relating to this cohort therefore may well be applicable to ethnic minority groups in the UK who originate from *Strongyloides*-endemic areas, such as those from Sub-Saharan Africa. A case may also exist for improving access to ivermectin in the primary care setting―although unlicensed, this has been proven to be a safe and effective treatment for infection with *Strongyloides* as a single dose [29]. A recent systematic review has also supported the role of screening for and treating strongyloidiasis in migrant populations and further evaluated the cost-effectiveness of presumptive single dose treatment with ivermectin [30].

In conclusion, *Strongyloides* is a highly prevalent chronic infection in primary care setting serving migrants from a highly endemic region, and eosinophilia was a clear predictor of infection in this context. Infection was also highly prevalent among patients attending the practice for gastrointestinal symptoms. Based on these data, there is likely an under-characterised, but significant burden of strongyloidiasis across the South Asian migrant population in London. This is an important infection has potential life-threatening consequences particularly in iatrogenic immunosuppression or malignancy, and highlights the importance of investigating the causes of eosinophilia. 

## Figures and Tables

**Figure 1 pathogens-09-00103-f001:**
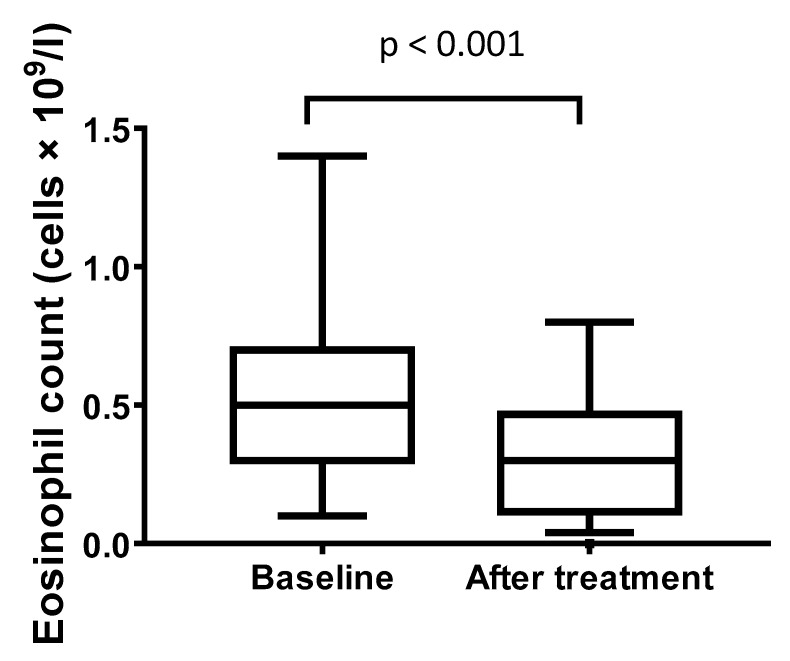
Box plot comparing eosinophil count at baseline, and after treatment. Horizontal line within the box represents the median and boundaries the 25th and 75th percentile. Whiskers indicate the 2.5 and 97.5% values with outliers represented as squares.

**Table 1 pathogens-09-00103-t001:** Baseline characteristics of the eosinophilia and gastrointestinal cohort with reference to the control phlebotomy group. OR: odds ratio; aOR: odds ratio adjusted for age, sex, time spent in the UK and country of origin; NS: non-significant difference taken as *p*-value > 0.05; * Patients recruited into the GI cohort who had an unexplained eosinophilia (17 patients) were included in both analyses.

	Control (Reference Cohort) (n = 120)	Eosinophilia Cohort (n = 223) *	*p*-Value	GI Cohort (n = 177)	*p*-Value
**Laboratory Parameters**					
*Strongyloides* serology Negative Positive	105 (87.5%) 15 (12.5%)	148 (66.4%) 75 (33.6%)	OR (95% CI): 3.55 (1.93–6.52), *p* < 0.001 aOR (95% CI): 3.54 (1.88–6.67), *p* < 0.001	148 (83.6%) 29 (16.4%)	OR (95% CI): 1.37 (0.70–2.68), *p* = 0.36 aOR (95% CI): 1.46 (0.74–2.89), *p* = 0.27
Median eosinophil count cells × 10^9^/L (IQR)	0.2 (0.1–0.3)	0.6 (0.5–0.7)	<0.001	0.2 (0.1–0.4)	0.008
Median serology titre for patients with positive serology (IQR)	1.2 (1.1–1.5)	1.6 (1.2–3.0)	0.03	1.3 (1.2–2.0)	NS
**Sociodemographic Variables**					
Median age in years (IQR)	51 (42–60)	47 (37–62)	NS	47 (37–60)	NS
Male sex	44 (36.7%)	127 (57.0%)	0.001	63 (35.8%)	NS
Bangladesh as country of birth	107 (89.2%)	193 (86.6%)	NS	161 (91.0%)	NS
Median number of years living in UK (IQR)	23 (14–32)	26 (15–37)	NS	23 (14–31)	NS
Median number of years since last travel to endemic area (IQR)	2 (1–5)	2 (1–5)	NS	3 (1–6)	NS
**Impact on Life**					
Median quality of life impact score (IQR) Mean quality of life impact score	0 (0–4) 10	0 (0–35) 19	0.001	36 (0–69) 39	<0.001
Median IBS score (IQR)	10 (0–104)	54 (0–160)	0.002	188 (73–291)	<0.001
Work missed No days missed Fewer than 30 days More than 30 days	102 (85.0%) 4 (3.3%) 3 (2.5%)	186 (83.4%) 28 (12.6%) 6 (2.7%)	0.03	105 (59.3%) 40 (22.6%) 26 (14.7%)	<0.001
**Medical History**					
History of GI investigations	36 (30.0%)	87 (39.0%)	NS	106 (60.9%)	<0.001
History of atopy	45 (37.5%)	100 (44.8%)	NS	83 (48.5%)	NS

**Table 2 pathogens-09-00103-t002:** Clinical characteristics between patients with a positive and negative *Strongyloides* serology. NS: non-significant difference taken as *p*-value > 0.05.

	*Strongyloides* Negative (n = 388)	*Strongyloides* Positive (n = 115)	*p*-Value
**Characteristics**
Male sex	163 (42.3%)	62 (53.9%)	0.03
Median age (IQR)	48 (37–61)	49 (42–59)	NS
Median eosinophil count (IQR)	0.4 (0.2–0.5)	0.5 (0.3–0.7)	<0.001
Median number of years in the UK (IQR)	24 (14–32)	26 (15–37)	NS
Bangladesh heritage	344 (88.7%)	102 (88.7%)	NS
No work days missed	294 (75.8%)	90 (78.3%)	NS
Underlying immunosuppression	1 (0.3%)	1 (0.9%)	NS
Median impact on quality of life score	0 (0–37)	0 (0–54)	NS
Median IBS Score	76 (0–200)	70 (0–181)	NS
**Symptoms**
Abdominal Pain	184 (47.4%)	46 (40.0%)	
Distension	145 (37.4%)	53 (46.1%)	
Vomiting	36 (9.3%)	9 (7.8%)	
Indigestion	172 (44.3%)	48 (41.7%)	
Diarrhoea	82 (21.1%)	19 (16.5%)	
Constipation	171 (44.1%)	51 (44.4%)	
Flatulence	162 (41.8%)	38 (33.0%)

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
