# Peer review of "High Prevalence of Strongyloides among South Asian Migrants in Primary Care―Associations with Eosinophilia and Gastrointestinal Symptoms"

_pathogens, 2020, doi:10.3390/pathogens9020103_

Round 1

Reviewer 1 Report

This is a well-written and important study.

Access to ivermectin for strongyloidiasis in primary care and general practice settings in the UK is imperative.

In Australia, Ivermectin has been approved for onchocerciasis and strongyloidiasis for 20 years.

Treating chronic strongyloidiasis is a primary health care role.

Author Response

Thank you for your comments – we agree primary care management of Strongyloidiasis, particularly in a high prevalence region such as East London, would improve patient care.

Yours sincerely,

Damien Ming and Mike Brown, on behalf of the authors

December 2019

Reviewer 2 Report

The Msc deals with the sero-prevalence of Strongyloides among south-asian migrants in London taking into account the eosinophilia, GI and life aspects.

I consider the study as incomplete and will explain later why.

The context is new but the datas are not.

Mat & Met should be divided in subsections.

Result section :

4.2. :

- I am curious to see the values for eosinophlia. I am not confortable with a « so » significant p value (<0.001) with a so few difference in the values. Moreover, the ranges (min/max ?) are overlapping.

- paragraph 2 : Does it refers to sero-prevalence ? I think so. Please clarify it.

- If I understood correctly : 35/115 sero+ had normal eosinophilia. From these 35, 11 gave stools (from which 1 was positive). It means than 80/ 115 sero+ had anormal eosinophilia.

Why no stools were taken to see Strongyloides ? Sero+ doesn’t mean that there is infection.

4.3. :

- Please also clarify « sero » positive

- For eosinophil counts, same remark than for 4.2. subsection

- Why 87 patients were treated whereas stools positivity was unknown ? Systematic treatment can lead to many resistance for different helminths and is not recommended

Discussion section :

Line 6 : I do not see evidence of Strongyloides infection in this Msc. Please explain it. Many paragraphs deals with no explanation or too much difficulties (line 30 : « unexplained eosinophilia », line 43 : « it is difficult to conclude », …) Paragraph 5 (from line 44 to line 56) deals with all the limitationsof the study. I agree with these limitations but I think, as authors were aware of them,they should have been fixed for the study to be complete. From a lot of patients, at the end only a few number went throug all the tests. This gives really less importance of the study. It is a pity that there is no discussion around the positive stool found in the « normal eosinophil count group » (see result 4.2., end of paragraph 1). I am a bit surprised that there is not more discussion about the immunological balance (Th1/Th2/Th17/Treg). In south eat Asia, a lot of other infectious diseases are very common. Other helminths infections but also protozoan (Leishmania, Plasmodium,….) This can counterbalance the cell dominance and other parameters.

Conclusion section :

« eosinophilia was a clear predictor of infection » should be removed. It is well kown thet hypereosinophilia occurs in almost all Helminths infection and is not specific to Strongyloides.

General remarks :

I am a bit surpised that no tentative for detection of other pathogens was realized to be sure that there is not interference with the Strongyloides infection For the negative stools, I suggest to take more timepoint to conclude to negative result as some low level infections could not be seen sometime.

Some spelling remarks :

Abstract :

Strongyloides (italic lease)

Mat & Met :

What does GP mean ? Please give the full spelling when used the first time. Statistical analysis should be a sub-section of Mat & Met Line 112 : please add « was » before « employed »

Discussion :

Line 31 : Strongyloides (italic please)

Author Response

Thank you,

Damien Ming and Mike Brown

Reviewer 3 Report

As a Reviewer, I would have the following remarks regarding the manuscript:

The inclusion and exclusion criteria should be improved.

The lot of patients is not enough, it should be extended. The sensitivity and specificity of the test  are low. The authors could not demonstrate an exact correlation between the eosinophilia and strongyloidiasis.

Considering all these comments, I suggest substantial changes in this manuscript in order to:

-        introduce more recent articles to the references;

-        please add more exclusion criteria;

-        please add some more conclusions;

I think this article needs major improvements to the experimental part and the batch size.

Thank you!

Author Response

Thank you for your comments – the inclusion, and exclusion criteria were deliberately broad to reflect the real-world nature of the study, in order for this to be applicable for use in the primary care setting in East London. We excluded patients on the basis of incomplete data or withdrawal from the study otherwise there were no other exclusion criteria. We agree that a larger number of patients would strengthen the study however this was not practical, nor feasible for a variety of reasons.

The relationship between eosinophilia and strongyloidiasis has been addressed in previous reviewers’ comments: please refer to the revised manuscript.

The references and dates are also a reflection of the state of research into strongyloidiasis and have been chosen as they are the most relevant. We have referenced a recent paper by Agbata et al., Int J Environ Res Public Health. 2019 Jan; 16(1): 11.) which supports the role of migrant screening for strongyloidiasis using serological methods and single-dose treatment (Lines 70-73).

We have phrased the discussion to focus on the clinical applications findings and have not addressed the immunological, nor haematological aspect of the results. We feel this would be beyond the scope of our article.

Yours sincerely,

Damien Ming and Mike Brown, on behalf of the authors

December 2019

Reviewer 4 Report

This is an exciting study and it is important to make aware the high prevalence of undiagnosed Strongyloides infection in a susceptible population and to treat this neglected pathogen, particularly to be mindful of this condition in those with eosinophilia and gastrointestinal symptoms, to perform serology. Treatment should be available in primary care – particularly given the resolution of symptoms following treatment.

However, there is a point I would take issue with and that is the classification of IBS - whilst not linked to individual bowel symptoms, the authors suggest that there is no link to IBS. This is a debatable point. The irritable bowel severity scoring system (IBSSS) dates back to 1997 and since this time the clinical diagnosis of IBS has changed considerably and under the Rome foundation consensus, IBS diagnosis relies almost exclusively on symptom patterns. Rome diagnostic criteria are the most widely accepted as the gold standard. The latest iteration is Longstreth GF, Thompson WG, Chey WD, et al. Functional bowel disorders. Gastroenterology 2006;130:1480-91.

The key is an association of pain with defecation and/or changes in stools and the different subtypes (which likely have differing underlying pathologies) are categorised by stool consistency. The IBSSS score is obtained by adding the scores together thus no separation of IBS type is obtained – the patients with Strongyloides may well have IBS- diarrhoea - but may have IBS-constipation or a mixed type IBS – this is not discernible from the analysis. IBS severity is not IBS diagnosis. IBS is common, and functional bowel disorders affect 1 in 4 adults in the US, Canada and the UK (Prevalence of Rome IV Functional Bowel Disorders among Adults in the United States, Canada, and the United Kingdom. Palsson, Olafur S. et al. Gastroenterology in press), finding a cause – post infectious IBS for example is important, as previously stated different types have likely different aetiologies. Of note in this study, Strongyloides antibodies among patients with GI symptoms were reasonably high (16.4%) . It would be worthwhile reclassifying the GI symptoms by IBS diagnostic criteria before excluding this pathogen as responsible for some IBS groups, particularly as a treatment exists. The fact that patients within the eosinophilia cohort also had a significantly lower reported quality of life and increased GI symptomatology when compared with controls does fit with IBS diagnosis.

The discussion needs reworking to take this into account – is it possible to define IBS from the questionnaires linking abdominal pain and stool consistency?

I would suggest getting a gastroenterologist on board as an author to advise on IBS definitions and analysis of questionnaires

Author Response

Thank you again for your comments - they have been indeed helpful in refining the manuscript.

The authors

Round 2

Reviewer 2 Report

I am still not convinced of the importance of this study.

I agree with only a little of the author's answers and changes.

But I think that in most of the questions, instead of answering, they just try to explain why they did what I questionned.

One important point, authors answer that there is no resistance to Ivermectin known up to date what is true in Humans but wrong for exemple in Animal Health (resistance for Parascaris equorum). Of course, if migrants are an isolated Community in London, there should be resistance. But, why giving ivermectine to the seropositive patients, if there is no confirmation of the presence of the parasite. I think it would be quicker and cheaper to give Ivermectin to all migrants and it is done.

I am sure it is important to estimate the rate of positive migrant coming in London, moreover because they come from an higly endemic region but not with this kind of study design.

Author Response

We thank you again for your time and your comments.

Indiscriminate use of ivermectin has the potential for driving resistance in helminths. Although there is no evidence of resistance in human helminth infection thus far, we agree with you that this does not mean its absence. As you have alluded to, a One Health approach for antimicrobial use, particularly the role of ivermectin in agriculture is important to consider. However, this is beyond the scope of our article.

Although presumptive treatment of ivermectin to migrants originating from Strongyloides-endemic regions would indeed be quicker and cheaper, our cohort is a heterogenous group, some of whom have been in the UK for many years. We therefore believe in the value of serological screening for targeted treatment – this approach will also reduce inappropriate ivermectin use and reduce the risk of resistance. The role of serology and microscopy has already been discussed in our previous response.

Reviewer 3 Report

Dear author, thank you for the opportunity to review the new version of the manuscript. In this form it can be accepted for publishing.

Best regards!

Author Response

Thank you for your kind and helpful comments.

Regards,

The authors

Reviewer 4 Report

Thank you for satisfactorily addressing my concerns